# Relationship between Flavonoid Chemical Structures and Their Antioxidant Capacity in Preventing Polycyclic Aromatic Hydrocarbons Formation in Heated Meat Model System

**DOI:** 10.3390/foods13071002

**Published:** 2024-03-25

**Authors:** Thi Thu Huong Huynh, Wanwisa Wongmaneepratip, Kanithaporn Vangnai

**Affiliations:** 1Department of Food Science and Technology, Faculty of Agro-Industry, Kasetsart University, Bangkok 10900, Thailand; huynhthithuhuong.h@ku.th; 2Department of Food Science and Technology, Faculty of Science and Technology, Thammasat University, Pathum Thani 12121, Thailand; wanwisaw@tu.ac.th

**Keywords:** polycyclic aromatic hydrocarbons, flavonoids, antioxidant activities, meat model system, DPPH, ABTS, FRAP

## Abstract

The relationship between the chemical structures of six flavonoids and their abilities to inhibit the formation of polycyclic aromatic hydrocarbons (PAHs) in a heated meat model system was investigated. The PAH8 forming in samples was analyzed by using QuEChERS coupled GC-MS. Inhibitory effects of PAHs were myricetin (72.1%) > morin (55.7%) > quercetin (57.3%) > kaempferol (49.9%) > rutin (32.7%) > taxifolin (30.2%). The antioxidant activities of these flavonoids, assessed through (1, 1-diphenyl-2-picrylhydrazyl) free radical scavenging activity assay (DPPH), [2,2′-azinobis (3-ethylbenzothiazoline-6-sulphonic acid)] free radical scavenging activity assay (ABTS) and ferric ion reducing antioxidant power assay (FRAP) assays, exhibited a significant negative correlation with PAH reduction. Notably, myricetin that contained three hydroxyl groups on the B-ring, along with a 2,3-double bond in conjugation with a 4-keto moiety on the C-ring, demonstrated strong antioxidant properties and free radical scavenging abilities, which significantly contributed to their ability to inhibit PAH formation. However, rutin and taxifolin, substituted at the C-3 position of the C-ring, decreased the PAH inhibitory activity. The ABTS assay proved the most effective in demonstrating the correlation between flavonoid antioxidant properties and their capacity to inhibit PAH formation in heated meat model systems. Thus, the inhibition of PAHs can be achieved by dietary flavonoids according to their chemical structures.

## 1. Introduction

Polycyclic aromatic hydrocarbons (PAHs), which are known carcinogens, are produced when organic substances undergo incomplete combustion or pyrolysis [1]. Benzo(a)pyrene (BaP), identified by the International Agency for Research on Cancer as a Group 1 carcinogen, is considered the most toxic PAH compound with a significant cancer risk. Additionally, for a more in-depth examination of the reasons behind the presence of toxic compounds in cooked meat, the focus is placed on the PAH8 group, consisting of benz(*a*)anthracene (BaA), chrysene (Chry), benzo(*b*)fluoranthene (BbF), benzo(*k*)fluoranthene(BkF), benzo(*a*)pyrene(BaP), indeno(*1*,*2*,*3-cd*)pyrene (InP), dibenz(*a*,*h*)anthracene (DahA) and benzo(*g*,*h*,*i*)perylene (BghiP) [2]. The content of PAHs in cooked meat depends on many factors, such as types of heating sources, cooking time and temperature, and cooking methods. It has been speculated that PAHs may form through the combination of free radicals, intramolecular reactions, or the polymerization of small compounds [1,2]. Due to the Maillard reaction and lipid oxidation, free radicals are produced during high-temperature food combustion [1]. These radicals combine to form light PAHs, and the latter combine with other radicals to form heavy PAHs, which are retained in food [2]. Furthermore, increased PAH formation in the heated meat model was reported to be related to the free radical oxidation of unsaturated fatty acids in the heated meat model system [1]. Preventing PAH formation in cooked meat is critical in the food industry to enhance safety.

Flavonoids are abundant polyphenols in plant tissues and fruits, making them a significant part of our daily diet. Flavonoids are composed of two aromatic rings linked by three carbons in an oxygenated heterocycle. According to the structure, dietary flavonoids could be subdivided into flavones, flavonols, flavanones, flavononols, flavonols, isoflavones, and anthocyanidins [3]. The arrangement of hydroxyl groups on the B-ring, their quantity, and the presence of a 2,3-double bond along with a 4-oxo group in the C-ring play pivotal roles in determining the antioxidant and free radical-scavenging properties of flavonoids [3]. Previous investigations have demonstrated successful inhibition of PAHs in cooked meat through the application of single flavonoids (e.g., quercetin) [4] or various herbs, spices, plant extracts abundant in flavonoids through a mechanism involving the scavenging of free radicals, such as garlic [4], onions [4], garlic essential oil, onion peel extract, ginger, and spice mixtures (cumin, coriander, and black pepper) [5]. Effective inhibition of PAHs in cooked meat has been proven to be related to applying ingredients with flavonoids with higher antioxidant activities [1]. The extent to which individual flavonoids can effectively inhibit the formation of PAHs remains uncertain. Examining the effectiveness of every flavonoid against PAHs is not feasible due to their vast numbers in nature. Nonetheless, there remains a lack of systematic processes in selecting flavonoids with potent PAH inhibitory activities. Therefore, establishing a systematic approach for selecting flavonoids and investigating their impact on PAH inhibition can offer a theoretical framework for effectively controlling PAH formation.

Previous research has extensively investigated the relationship between the chemical structures of flavonoids and a wide range of pharmacological and biological activities, including antioxidant activity, anti-cancer, anti-inflammatory, anti-diabetic, and enzyme inhibition effects [6,7]. This is a reliable tool for predicting the properties of flavonoids at minimal costs and providing a better understanding of the inhibition mechanism involved. For example, the relationship of flavonoids’ chemical structure has been successfully studied for predicting their inhibitory effects on heterocyclic aromatic amines, another carcinogenic compound found in cooked meat, revealing the importance of hydroxyl groups and specific topological structures for inhibitory effects [8]. To our knowledge, the structure–activity relationship of flavonoids, specifically in PAH inhibition, has not been studied. Thus, the major objective of this study was to explore the relationship between the structure of each flavonoid and its ability to inhibit PAHs.

In this study, six flavonoids, including kaempferol, myricetin, quercetin, morin, rutin, and taxifolin, have been selected based on their different chemical structures regarding the number of hydroxyl groups on B-ring, the presence of C_2_ = C_3_ on C-ring, and the glycoside form. The molecular structure, including the subclasses of all six tested flavonoids, is shown in Table 1. The investigation was conducted in a controlled heated meat model system to eliminate the inherent variability in meat composition and cooking conditions, allowing for the investigation of the specific impact of added flavonoids on PAH formation. The outcomes of our investigation hold promise for the food industry, offering insights into product formulation strategies that integrate flavonoid-rich ingredients known for their potent PAH inhibitory properties, thereby mitigating potential health risks associated with PAH consumption.

## 2. Materials and Methods

### 2.1. Materials, Chemicals and Reagents

Ground beef from beef chuck (7 g fat/100 g meat) was purchased from a local market in Bangkok, Thailand. Six flavonoids, including kaempferol, myricetin, quercetin, morin, rutin, and taxifolin were purchased from Sigma (St. Louise, MO, USA). We obtained 2,2-diphenyl-1picrylhydrazyl (DPPH), 2,2′-azino-bis-(3-ethylbenzothiazoline-6-sulfonate) (ABTS), potassium persulfate, 37% hydrochloric and sodium acetate acid from Aldrich (Steinheim, Germany). TPTZ (2,4,6-tri (2-picridyl)-triazine, Iron (III) chloride hexahydrate, Iron (II) Sulfate Heptahydrate of Grade AR were purchased from QRec (Auckland, New Zealand). 

EPA PAH8: (benz(*a*)anthracene (BaA), chrysene (Chry), benzo(*b*)fluoranthene (BbF), benzo(*k*)fluoranthene (BkF), benzo(*a*)pyrene (BaP), indeno(*1*,*2*,*3-cd*)pyrene (InP), dibenz(*a*,*h*)anthracene (DahA) and benzo(*g*,*h*,*i*)perylene (BghiP)) and QuEChERS kits were purchased from Supelco (Bellefonte, PA, USA). Internal standards (Acenaphthene-d10 and Chrysene-d12) were purchased from Dr. Ehrenstorfer (Augsburg, Germany), magnesium sulfate and sodium acetate were purchased from Ajax Finechem (Silverwater, NSW, Australia), all solvent high-performance liquid chromatography (HPLC) grade—(acetonitrile, methanol), ethanol (AR grade) and n-hexane (AR grade) were obtained from RCI Labscan (Bangkok, Thailand).

### 2.2. Antioxidant Activities of Six Flavonoids

#### 2.2.1. Sample Preparation

Each tested flavonoid was dissolved in ethanol at a concentration of 100 mg L^−1^. Subsequently, dilutions with ethanol were made to achieve concentrations of 0, 3, 5, 10, 15, 20, 30, and 35 mg L^−1^ for each flavonoid. The antioxidant activities of each diluted flavonoid were then determined as described below.

#### 2.2.2. DPPH Scavenging Activity

The free radical scavenging activity of six tested flavonoids was determined using DPPH reagent through a method described by Zhao et al. (2020) [9]. Each tested flavonoid (2 mL) was added to 2 mL of DPPH solution to initiate the reaction, obtaining a calibration curve. The absorbance of the mixture was read at 517 nm using a UV-VIS spectrophotometer (Model: Genesys 10-S, Thermo Fisher Scientific Inc, Waltham, MA, USA). DPPH solution was used as the control, and ethanol was used as the blank. The result was revealed as EC_50_ (mg/mL) of DPPH scavenging activity by observing the 50% inhibitory concentration for each tested flavonoid using the calibration curve.

#### 2.2.3. ABTS Assay

ABTS radical scavenging activity was determined following the method described by Zhao et al. (2020) [9] with some modifications. The ABTS assay is based on the scavenging of the 2,2′-azino-bis-(3-ethylbenzothiazoline-6-sulfonate) radical (ABTS•+). ABTS solution was produced by reacting 7 mM ABTS solution with 2.45 mM potassium persulfate solution (1:1) (*v*/*v*) and storing the mixture in the dark at room temperature for 16 h before use (color changed from light blue to dark blue). The absorbance of the ABTS solution was measured and adjusted to 0.700 ± 0.025 at 734 nm using a UV/VIS spectrophotometer. Then, 200 μL of tested flavonoid was mixed with 1 mL of ABTS solution and left at room temperature in the dark for 6 min. The absorbance of the mixture was read at 734 nm using a UV-VIS spectrophotometer. Ethanol was used as the blank. The result was revealed as EC_50_ (mg ml^−1^) of ABTS scavenging activity by observing each tested flavonoid 50% inhibitory concentration using the calibration curve.

#### 2.2.4. FRAP Assay

The FRAP assay was conducted according to previous methods with some modifications [10,11]. This assay’s principle is based on reducing a ferric-tripyridyltriazine complex to its ferrous-colored form in the presence of antioxidative compounds. The FRAP reagent was freshly prepared by combining 2.5 mL of a 10 mM TPTZ (2, 4, 6 tripyridyl-s-triazine) solution in 40 mM HCl plus 2.5 mL of 20 mM FeCl_3_.6H_2_O and 25 mL of 300 mM acetate buffer (pH 3.6). First, the absorbance of fresh FRAP reagent was read at 593 nm using a UV-VIS spectrophotometer. Then, 40 μL of tested flavonoid was mixed with 200 μL distilled water and 1850 μL FRAP reagent, and the absorbance of the reaction mixture at 593 nm was measured. Finally, the absorbance was measured again after 30 min incubation in the dark. Ethanol was used as the blank sample. The FRAP values were achieved by a standard calibration curve obtained using different concentrations of the FeSO_4_ solution (0.1–2 mM). The final result was expressed as the concentration of tested flavonoid having a ferric ability equivalent to that of 1 mM FeSO_4_ (equivalent concentration 1 = EC_1_).

### 2.3. Inhibitory Effects of Flavonoids on PAH Formation in Heated Meat Model System

The lyophilization of ground beef was prepared using a freeze dryer (FDA-8508; Ilshin Bio Base Co., Ltd., Dongducheon-si, Korea). The freeze-dried meat was homogenized and stored at −20 °C until used for experiments. The meat model system was designed using a slightly modified method described by reports [12,13]. All six tested flavonoids were prepared in ethanol. In total, 1 g lyophilized ground beef was put directly into glass test tubes, and 1 mL of tested flavonoid was added (250 µg kg^−1^). The control sample was prepared by adding 1 mL of ethanol instead of tested flavonoids, and then tubes were sealed with caps. The experiment was conducted in a fume hood. The oil bath was heated at 180 °C before heating the samples. The glass test tubes containing samples were heated at 180 ± 5 °C for 20 min. After finishing heating, test tubes were immediately cooled in an ice bath to interrupt the reaction. The tubes were kept at −20 °C until being used for analysis.

### 2.4. PAH Extraction and Clean-Up

The extraction and purification methods were modified in the previous study [13]. The heated meat samples were removed from the tubes and thawed at 4 °C overnight. A portion of 1 g of the sample was transferred to a 50 mL centrifuge tube and homogenized in 10 mL of deionized water for 1 min. The mixture was spiked with the internal standard 200 ng/g (Chrysene-d12), followed by adding 10 mL of acetonitrile and vigorous shaking for 1 min by hand. After 15 min, 6 g of MgSO_4_ and 1.5 g of CH_3_COONa were added, shaken for 1 min and centrifuged at 4000 rpm for 5 min at 25 °C. Subsequently, 6 mL of the supernatant was mixed with a QuEChERS clean-up tube containing 400 mg of PSA, 1200 mg of MgSO_4_, and 400 mg of C18EC for purification, followed by centrifuging at 4000 rpm for 5 min at 25 °C. The supernatant was collected and purged using N_2_ gas to dry and then dissolved in 100 μL of acetonitrile and a 1 μL aliquot was for injection into GC-MS for PAH analysis.

### 2.5. PAH Analysis

The sample extracts were analyzed using Gas Chromatography (7890B, Agilent, Mississauga, ON, Canada) and detected using MS (GC-MS) (5977A, Agilent, Canada). The separation of PAHs was performed on an Agilent DB-5 MS column 30 m × 0.32 mm × 0.25 μm, J&W (Scientific, Folsom, CA, USA) following Lynam and Smith (2008), and the carrier gas is 99.999% helium (1.5 mL min^−1^). The injection mode is split-less (300 °C, 1 μL). The oven temperature was programmed at 55 °C for 1 min, ramped at 25 °C min^−1^ to 320 °C and held for 3 min. The transfer temperature was set at 330 °C. All analyses were performed in the selected ion monitoring mode (SIM) of the MS to enhance the selectivity and sensitivity of the method. PAH8 was quantified using the internal standard method, and the target ions monitored for analyses are 228, 240 (IS), 228, 252, 276, and 278 for the PAH8 and the deuterated internal standard solution (Chrysene-d12). Six-point calibration curves (5–500 ng ml^−1^) for each PAH were used to quantify the amount of each PAH in samples. All eight PAHs were quantified using the relative response factors related to internal standards.

The limits of detection (LODs) and quantification (LOQs) were calculated using the signal-to-noise ratios of S/N = 3 and S/N = 10, respectively [14]. PAH recoveries were carried out by spiking 200 ng g^−1^ of PAH8 standard mixtures into the sample. An unspiked sample was used as the control. Recovery of each PAH was calculated through the differences obtained in total concentrations between both spiked and unspiked samples [15].

### 2.6. Experimental Plan and Statistical Analysis

The experimental design, a completely randomized design, was used in the whole study. All experiments were conducted in triplicate. Differences in means were determined in Duncan’s multiple range test using the application of SPSS version 19.0 (IBM SPSS, Chicago, IL, USA) with a 95% confidence level (*p* < 0.05). Pearson’s correlation coefficient was calculated between differential antioxidant activities and PAH inhibitory effects of tested flavonoids.

## 3. Results and Discussions

### 3.1. Validation of PAH Analysis Method

The eight PAHs could be detected, and the chromatogram of the standards is shown in Figure 1. The retention time, LODs, LOQs, R^2^, and recoveries are shown in Table 2. The limits of detection (LODs) and limits of quantification (LOQs) for PAH8 varied from 0.01 to 0.05 µg kg^−1^ and 0.09 to 0.15 µg kg^−1^, respectively. The squared correlation coefficients of determination (R^2^) of the calibration curve were found to be over 0.9900. The recovery rates fell within the range of 70.1% to 109.4%. However, Chry and B*k*F exhibited lower recovery rates due to partial loss during the extraction and clean-up process, as reported by [16]. Adhering to the guidelines of [17] EC No. 835/2011, which stipulates a recovery range of 50–120%, the recovery rates observed in this study were acceptable.

### 3.2. Inhibitory Effects of Various Antioxidants on PAH Formation in Meat Model System

The structure–activity relationship of flavonoids, specifically in PAH inhibition, was investigated in this study. Six different flavonoids, including kaempferol, myricetin, quercetin, morin, rutin, and taxifolin had been selected due to their unique chemical structures in these three criteria: (1) hydroxylation patterns on B-ring including the different number of hydroxyl groups on B-ring [kaempferol (4′-OH), quercetin (3′,4′-OH), and myricetin (3′,4′,5′-OH)], and different pattern of hydroxyl group [quercetin (ortho position), morin (meta position)], (2) presence of C_2_ = C_3_ double bond on C-ring [quercetin (presence of C_2_ = C_3_), taxifolin (lack of C_2_ = C_3_)], and (3) glycoside form [quercetin (aglycone), rutin (glycoside form of quercetin)].

The antioxidant potential of the flavonoids was assessed through DPPH, ABTS, and FRAP assays, providing respective EC_50_ DPPH, EC_50_ ABTS, and EC_1_ FRAP values. The DPPH and ABTS assays are based on the scavenging activity toward a stable free radical, while the FRAP assay focuses on reducing metal ions [18]. To investigate the PAH inhibitory effects of flavonoids, the percentage of PAH inhibition was calculated by comparing the PAH contents of each model and control model (no added flavonoid). Seven PAHs were detected for the control model, including Chry, B*b*F, B*k*F, B*a*P, I*n*P, D*ah*A, and B*ghi*P. BaA was not detected in any tested model. The PAH contents formed in the heated meat model were comparable with the previous studies conducted in the heated meat model [12]. The PAH inhibition effects of flavonoids were considered based on their chemical structures and antioxidant activities.

Table 3 presents the PAH8 levels in a meat model system. In comparison to the control sample, which involved the lyophilization of ground beef alone, the addition of kaempferol, morin, myricetin, and quercetin resulted in the inhibition of PAH8 generation by 49.85%, 55.7%, 72.14%, and 57.25%, respectively. The levels of BaP in samples treated with kaempferol, morin, myricetin, and quercetin were significantly lower than the control (2.44 μg kg^−1^), with myricetin exhibiting the lowest value (*p* < 0.05). While the four flavonoids demonstrated variations in antioxidant activity, the inhibition of PAH8 formation did not show significant differences (*p* < 0.05).

#### 3.2.1. Hydroxylation Patterns on B-Ring

The flavonols, kaempferol (4′-OH), quercetin (3′,4′-OH), and myricetin (3′,4′,5′-OH) had the similar structures for A and C rings but had different numbers of hydroxyl groups on the B-ring. All three antioxidant assays confirmed that quercetin had stronger antioxidant activities than kaempferol, and myricetin showed stronger antioxidant activities than kaempferol, indicating that antioxidant activities were increased with a higher number of hydroxyl groups, which was agreed with previous studies [3]. As shown in Table 3, the addition of kaempferol, quercetin, and myricetin led to a reduction in total PAH8 by 49.9%, 57.3%, and 72.1%, respectively. Notably, BaP and DahA were among the PAHs that exhibited the highest level of inhibition. The pattern of hydroxyl groups on B-ring was also investigated by comparing the inhibition effects of PAHs between quercetin and morin. Morin serves as an isomeric form of quercetin, sharing similarities having a hydroxyl group in position 3, a resorcinol moiety, and a carbonyl group in position 4; the only difference between them is the hydroxylation pattern on B-ring, which is *meta* in morin but *ortho* in quercetin [3]. Quercetin showed higher antioxidant activities in all three assays. However, the PAH inhibition effects of these two compounds were not significantly different (*p* ≥ 0.05).

Our results indicated that while both the quantity and positioning of hydroxyl groups on the B-ring enhanced the antioxidant activity of flavonoids, the inhibition effects on PAHs were primarily influenced by the number of hydroxyl groups on the B-ring. Previous research highlighted the significance of the number of hydroxyl groups, as they contribute to the exceptional stability of the aroxyl radical through mechanisms such as hydrogen bonding or expanded electron delocalization [19]. Through its engagement in electron delocalization, the hydroxyl group on the B-ring stabilizes flavonoid phenoxyl radicals, thereby enhancing their antioxidative potential [20].

#### 3.2.2. Presence of 2,3 Double Bond at C-Ring

Quercetin (presence of C_2_ = C_3_), characterized by a 4-keto group and 3-hydroxyl moiety, exhibited higher DPPH, ABTS, and EC_1_ FRAP EC_50_ values compared to taxifolin (lack of C_2_ = C_3_), as depicted in Figure 2 (*p* < 0.05). Conversely, Taxifolin possessed a B-ring catechol group but lacked the 2,3-double bond, resulting in significantly lower antioxidant activity than quercetin. This was evident in EC_50_ values of 29.68 mg L^−1^ for DPPH, 17.68 mg L^−1^ for ABTS, and 0.39 mM for FRAP. The degree of unsaturation in the C-ring had a notable impact on the antioxidant activity of flavonoids. The presence of the 2,3-double bond, coupled with the 4-keto group, enhanced the functionality of both the A and C-rings, facilitated electron donation from the 3,5-dihydroxyl groups, moderately increased the EC_50_ and EC_1_ values, delocalized electrons from the B-ring, and significantly reduced cellular antioxidant activity [21,22]. Additionally, the inhibition of PAHs correlated with the antioxidant activities of quercetin and taxifolin; the PAH8 content in the sample with added quercetin was significantly lower than in the sample with taxifolin, with values ranging from 18.29 µg kg^−1^ to 29.86 µg kg^−1^ (*p* < 0.05). These findings suggest that taxifolin’s diminished ability to inhibit PAH formation is due to the absence of the 2,3-double bond in conjunction with a 4-oxo function in the C-ring, which leads to lower antioxidant capabilities.

#### 3.2.3. Presence of 3-Glycoside Brand in C-Ring

Most dietary flavonoids exist in the form of glycosides in plants. The glycosylation could improve its solubility [3]. Quercetin, which lacks 3-glycosylation, demonstrated higher antioxidant activity than rutin, containing glycosylation at the C-ring, in both EC_50_ (DPPH and ABTS assays) and EC_1_ (FRAP assay). In terms of B*a*P levels, rutin samples (2.35 µg kg^−1^) exhibited much higher values than quercetin samples (0.12 µg kg^−1^), indicating that rutin did not reduce BaP compared to the control (2.44 µg kg^−1^). Furthermore, the inhibition of PAH8 formation in rutin samples was significantly lower (33%) compared to quercetin samples (57%) (*p* < 0.05). This discrepancy can be explained by quercetin’s possession of 3,5-dihydroxyl groups with a 4-oxo function in both the A and C-rings, which are essential for maximum free radical scavenging potential compared to rutin. Additionally, the presence of a 3-glycosyl group in flavonoids may increase the molecular volume and hinder access to the active free radical scavenging site [13], leading to a significant decrease in PAH inhibition capability. Our results revealed that glycosylation significantly weakened PAH inhibitory effects. This could be due to the glycosylation reducing the reactivity for free radical capture. The formation of intermolecular hydrogen bonds significantly hinders the dissociation of the H atoms on the phenolic hydroxyl group and the capture of excess free radicals. The significant impact of 3-glycosylation on the antioxidant activity of rutin is attributed to the absence of hydrogen donors due to this glycosyl group [9].

Overall, myricetin-treated samples had a substantially higher PAH8 inhibition (72.14%) than those with kaempferol, morin, and quercetin (49.85%, 55.7%, and 57.25%). Flavonoids prevented free radicals and light PAHs (BaA and Chry) from forming heavy PAHs, inhibiting most heavy PAHs (B*b*F, B*k*F, B*a*P, I*n*P, D*ah*A, and B*ghi*P) compared to control [6]. These results showed that flavonoids’ free radical scavenging activity to suppress heavy PAH8 production depended more on hydroxyl group numbers than B-ring position. Myricetin emerged as the most promising flavonoid for preventing PAH formation in heated meat models. By integrating myricetin-rich components (such as cranberries, grapes, and red wine) into recipes and promoting these products as antioxidant-rich, healthier alternatives, the food industry can offer consumers choices that reduce PAH formation during cooking, potentially enhancing both product appeal and health consciousness in the market [3].

### 3.3. Correlation between PAHs Reduction and Flavonoid Activity Accessed by EC_50_ (DPPH, ABTS Assay) and EC_1_ FRAP Assay

Figure 2 depicts the correlation between flavonoid-induced PAH inhibition and antioxidant activity (measured by EC50 values in DPPH, ABTS assays, and EC_1_ FRAP assay). In addition to Figure 2, it was shown that the lower EC_50_ values of DPPH, ABTS assays and EC_1_ of FRAP assay, the higher percentage BaP, PAH8 inhibition in the heated meat model system. The percentage inhibition of BaP was observed to have a negative correlation with EC_50_ ABTS (R^2^= −0.7670; *p* < 0.01). Additionally, inhibiting PAH8 formation showed negative correlations with both EC_50_ DPPH (R^2^= −0.7445; *p* < 0.01) and EC_50_ ABTS (R^2^= −0.7312; *p* < 0.01). This suggested that the antioxidant activity of flavonoids is inversely related to their ability to inhibit PAH formation. The ABTS assay was significantly negatively correlated with both BaP and PAH8 inhibition percentages, indicating that PAH inhibition primarily operates through free radical scavenging in heated meat models.

The findings indicate that the ABTS assay is the most effective method for assessing the relationship between flavonoid antioxidant activity and PAH inhibition in heated meat models. The EC_50_ value in the ABTS assay showed the strongest correlation (R^2^ = 0.5386 for % PAH8 inhibition and R^2^ = 0.5912 for % BaP inhibition), followed by the EC_50_ value in the DPPH assay (R^2^ = 0.5568 for %PAH8 inhibition). This suggests that PAH formation likely involves free radicals and can be hindered by antioxidants due to their radical-scavenging abilities. In other words, as determined by the ABTS method, flavonoids with higher antioxidant potential tend to exhibit a greater ability to inhibit PAH formation.

## 4. Conclusions

Flavonoids have shown significant potential in inhibiting the formation of PAHs when they possess specific characteristics: multiple hydroxyl groups on the B-ring, a 2,3-double bond in conjunction with a 4-keto moiety on the C-ring, and the absence of a glycoside group on the C ring. Among the methods employed, the ABTS assay proved most effective in demonstrating the relationship between flavonoids’ antioxidant properties and their capacity to inhibit PAH formation. This study provides a theoretical basis for controlling the PAH formation in meat-based products according to the structure of flavonoids.

## Figures and Tables

**Figure 1 foods-13-01002-f001:**
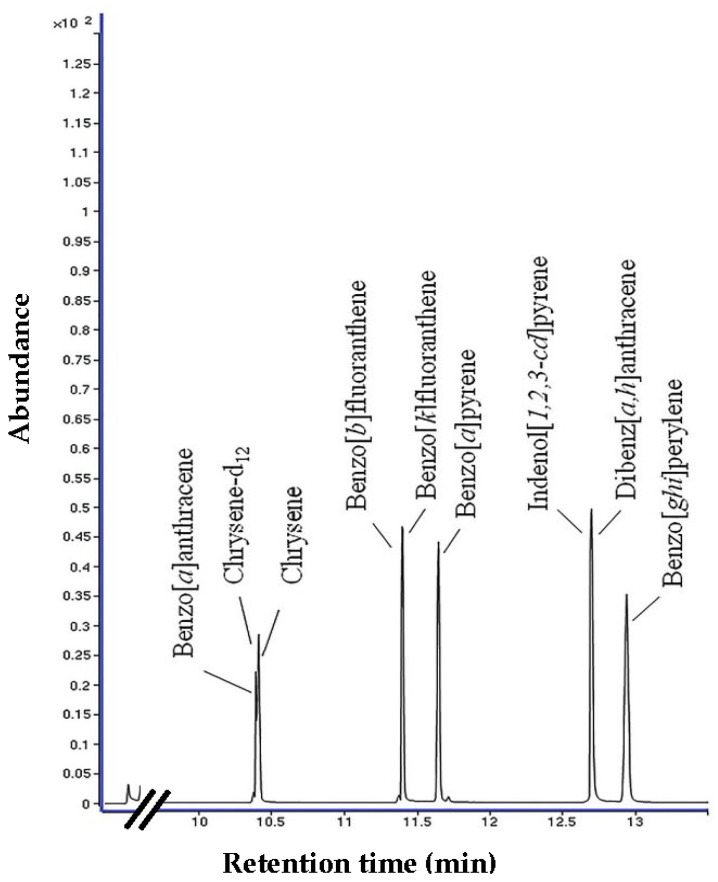
GC–MS chromatogram of PAH8 standard at 1 ppm (1 µL injection) using SIM mode.

**Figure 2 foods-13-01002-f002:**
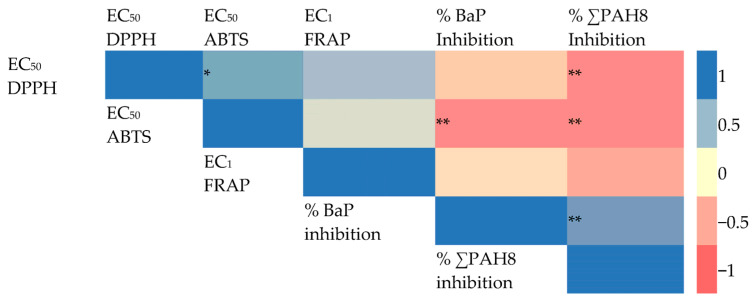
Pearson’s correlation coefficients of antioxidant activities of flavonoids accessed by DPPH, ABTS and Frap methods and PAH inhibition. * Correlation is significant at the 0.05 level (2-tailed). ** Correlation is significant at the 0.01 level (2-tailed). ∑ PAH8: BaA; Chry; BbF; BkF; BaP; InP; DahA and BghiP.

**Table 1 foods-13-01002-t001:** The molecular structures of six tested flavonoids.

Flavonoid	Class	Chemical Structure	Structural Formula								Molecular Formula
			R_3_	RÍ_5_	R_7_	R_2′_	R_3′_	R_4′_	R_5′_	Presence of C_2_ = C_3_	
Kaempferol	Flavonol	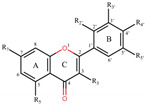 Flavonol	OH	OH	OH	H	H	OH	H	/	C_15_ H_10_O_6_
Quercetin	Flavonol		OH	OH	OH	H	OH	OH	H	/	C_15_ H_10_O_7_
Morin	Flavonol		OH	OH	OH	OH	H	OH	H	/	C_15_ H_10_O_7_
Myricetin	Flavonol		OH	OH	OH	H	OH	OH	OH	/	C_15_ H_10_O_8_
Rutin	Flavonol glycoside		Ogl	OH	OH	H	OH	OH	H	/	C_276_H_30_ O_16_
Taxifolin	Flavanonol	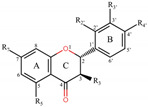 Flavanonol	OH	OH	OH	H	OH	OH	H	-	C_15_ H_12_O_7_

Source: [3].

**Table 2 foods-13-01002-t002:** The LODs, LOQs, R^2^ and recovery values obtained for eight PAH standards.

PAHs	Abbv.	Retention Time (min)	LODs (µg/kg)	LOQs (µg/kg)	R^2^	Recovery (%)
Benz[*a*]anthracene	B*a*A	10.39	0.03	0.09	0.9952	109.4
Chrysene	Chry	10.41	0.03	0.12	0.9860	84.5
Benzo[*b*]fluoranthene	B*b*F	11.39	0.05	0.15	0.9972	105.4
Benzo[*k*]fluoranthene	B*k*F	11.40	0.05	0.15	0.9990	70.1
Benzo[*a*]pyrene	B*a*P	11.65	0.05	0.15	0.9954	102.5
Indeno[*1*,*2*,*3-cd*]pyrene	I*n*P	12.69	0.01	0.12	0.9942	106.7
Dibenz[*a*,*h*]anthracene	D*ah*A	12.70	0.01	0.15	0.9918	101.6
Benzo[*g*,*h*,*i*]perylene	B*ghi*P	12.94	0.01	0.15	0.9990	90.7

**Table 3 foods-13-01002-t003:** Antioxidant activities and PAH inhibitory effects of six tested flavonoids.

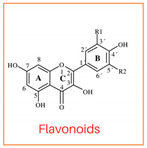		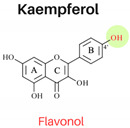		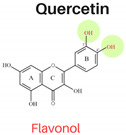		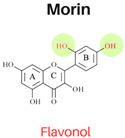		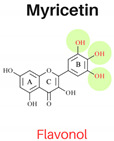		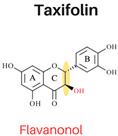		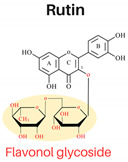	
**Antioxidant activities**													
DPPH EC_50_ (mg mL^−1^)		27.63 ± 0.52 b		18.43 ± 0.57 c		27.14 ± 0.82 b		13.77 ± 0.52 d		29.68 ± 0.05 a		26.05 ± 1.45 b	
ABTS EC_50_ (mg mL^−1^)		17.11 ± 0.12 b		7.41 ± 0.16 d		14.89 ± 0.63 c		5.78 ± 0.31 c		17.68 ± 0.01 b		33.46 ± 0.54 a	
FRAP EC1 (mM)		0.27 ± 0.002 b		0.16 ± 0.007 c		0.23 ± 0.001 e		0.15 ± 0.004 e		0.39 ± 0.012 d		0.22 ± 0.007 a	
**PAH inhibitory effects**													
**PAH**	**Control** **(μg kg^−1^)**	**μg** **kg^−1^**	**Inhibition**	**μg** **kg^−1^**	**Inhibition**	**μg** **kg^−1^**	**Inhibition**	**μg** **kg^−1^**	**Inhibition**	**μg** **kg^−1^**	**Inhibition**	**μg** **kg^−1^**	**Inhibition**
B*a*A	ND	ND	-	ND	-	ND	-	ND	-	ND	-	ND	-
Chry	7.26 ± 1.05 ab	5.38 ± 0.01 ab	25.9%	9.39 ± 0.99 ab	−29.3%	9.06 ± 1.22 ab	−24.8%	5.48 ± 0.93 b	24.5%	11.84 ± 2.99 a	−63.1%	11.44 ± 3.19 a	−57.5%
B*b*F *	1.78 ± 2.52	1.57 ± 0.31	11.9%	ND	100%	ND	100%	0.57 ± 0.26	68.2%	0.66 ± 0.01	62.9%	0.43 ± 0.61	75.8%
B*k*F	6.92 ± 0.37 a	1.83 ± 0.71 bc	73.6%	0.82 ± 0.24 b	88.1%	0.39 ± 0.09 c	94.3%	0.14 ± 0.20 c	98.0%	2.73 ± 1.78 bc	60.6%	4.00 ± 2.72 ab	42.2%
B*a*P	2.44 ± 0.56 a	0.18 ± 0.12 b	92.4%	0.12 ± 0.05 b	94.9%	0.15 ± 0.04 b	93.7%	0.06 ± 0.09 b	97.5%	1.14 ± 1.19 ab	53.1%	2.35 ± 1.17 a	3.47%
I*n*P *	0.46 ± 0.05	0.16 ± 0.07	65.3%	0.18 ± 0.27	61.5%	0.47 ± 0.67	−3.8%	ND	100%	0.68 ± 0.13	−48.2%	0.38 ± 0.14	17.7%
D*ah*A	23.26 ± 2.46 a	12.26 ± 2.30 b	47.3%	7.57 ± 0.71 b	67.5%	8.58 ± 0.20 b	63.1%	5.48 ± 2.40 b	76.4%	12.42 ± 1.51 b	46.6%	9.82 ± 3.2 b	57.8%
B*ghi*P	0.67 ± 0.09 a	0.08 ± 0.07 d	87.4%	0.21 ± 0.00 cd	69.5%	0.29 ± 0.06 bc	56.8%	0.19 ± 0.03 cd	72.3%	0.38 ± 0.08 b	42.7%	0.37 ± 0.02 b	45.1%
**Total PAH8**	42.79 ± 0.77 a	21.46 ± 1.90 bcd	49.9%	18.29 ± 0.82 d	57.3%	18.95 ± 2.10 cd	55.7%	11.92 ± 1.39 d	72.1%	29.86 ± 7.69 b	30.2%	28.79 ± 3.13 bc	32.7%

a–e Different letters indicate significant differences between mean values of each PAH (*p* < 0.05). * Indicates not significant (*p* ≥ 0.05).

## Data Availability

The original contributions presented in the study are included in the article, further inquiries can be directed to the corresponding author.

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
