# Peer review of "Relationship between Flavonoid Chemical Structures and Their Antioxidant Capacity in Preventing Polycyclic Aromatic Hydrocarbons Formation in Heated Meat Model System"

_foods, 2024, doi:10.3390/foods13071002_

Round 1
Reviewer 1 Report
Comments and Suggestions for Authors
Title:
· Do not abbreviate PAH
Abstract:
· Do not Abbreviate DPPH, ABTS, and FRAP
· The abstract should be formatted as Aim/purpose of the study, methodology, results, and conclusion.
· In the abstract, the GC-MS analysis was not indicated and should therefore be added.
· The results of the meat model system were not indicated as well.
· Since you only used 6 flavonoids, specify the identify of the flavonoids in the results part of the abstract
Keywords:
· DPPH, ABTS, and FRAP should be included in the keywords
Introduction:
· Page 1, Line 32. Do not abbreviate the enumerated PAHs since it was the first time you mentioned it in the manuscript
· Page 1, Line 37. [3] should be changed to the name of the author et al of the research and the citation will be placed at the end of the sentence.
· A justification on why the flavonoids used in the study were chosen can strengthen and improve the introduction.
· Cite and discuss some research about the antioxidant activity of flavonoids or phenolics against PAHs or other related carcinogens and explain why this research is unique.
· The aim of the study did not specifically indicate why the experiments were carried out and why they were chosen in the study. Improve this.
· The figure description of Figure 1 does not agree with the images that you included in Figure 1. The Fugue 1 shows the specific structures of the flavonoids used in the study and not the types or classes of flavonoids
· Figure 1. The names of the compounds should be placed below the structures.
· Figure 1. Taxifolin’s name is incomplete and should be adjusted properly.
· Figure 1. Generate a figure with better resolution.
· Figure 1. The information on the bottom part of the figure is incomplete. You have shown that each compound differs in substituents but you did not specify in which carbon number are they connected. Improve this and better check out other papers that report on flavonoid structures and learn from the way they represent.
· Figure 1. What does + and – mean?
· Figure 1. Indicate if how these structures were obtained or generated in the description of Figure 1
Materials and Methods:
· Section 2.1. Materials should be Materials, Chemicals and Reagents
· Start Section 2.1. with the description of how the ground beef was obtained.
· Add state for each supplier
· Indicate how the flavonoid solutions were prepared and which solvent was used to prepare them. Also, indicate the concentrations used.
· Section 2.2.1. 200 μL should be two hundred microliters. Do not start a sentence with a number. Check the manuscript and revise it.
· Section 2.2.1. without light should be in the dark or away from light
· Section 2.2.1. Specify the equipment and the supplier of the instrument used in measuring the absorbance.
· Section 2.2.1. Indicate the blank composition of the blank or control.
· Section 2.2.1. were the results of the DPPH assay first expressed as % inhibition, and then from it, the EC50 was calculated? If yes, incorporate the calculation of the % inhibition in the paper.
· Section 2.2.1. what software did you use to calculate the EC50?
· Section 2.2.2. Specify the name of the author from which the method was adopted
· Section 2.2.2. Indicate the instrumentation.
· Section 2.2.2. Indicate whether results obtained from the assay were expressed as % inhibition first
· Section 2.2.2. Indicate the blank composition of the blank or control.
· Section 2.2.3. Revise the Paragraphs and improve the English and the cohesiveness of the statements.
· Section 2.2.3. Indicate the way the flavonoid solutions were prepared and at which concentration. Also indicate the instrumentation and how the results were expressed.
· Section 2.2.3. Did you use FeSO4 in the FRAP assay? If yes, indicate the manner and the concentration on how it was prepared.
· Section 2.3. Rethink your title for Section 2.3. Your experiment is about the inhibition of the formation of the PAHs and not grounding meat.
· Section 2.4. Were the experiments performed in triplicate?
· Section 2.5. the numbers asfter the chemical symbols should be in subscript form.
· Section 2.6. What was the ionization mode of the MS?
· Section 2.6. Standard PAHS used should be enumerated here as well as the concentrations on how it was prepared.
· Section 2.7. P should be p and should be applied all throughout the manuscript. Also, you performed a correlation, therefore you should include it here in the methodology and statistical analysis.
Results and Discussion
· Section 3.1. Radical Scavenging mechanism of DPPH and ABTS can also be in a form or electron transfer. Improve this.
· Figure 2. X-Axis Should be DPPH (EC50 in mg/L), ABTS (EC%0 in?), and FRAP (EC1 in mM FeSO4). Fix the Title of the Image. Also, the pattern of the figures hides the error bars of some graphs. Fix this and probably use black and white shades instead.
· Since Figure 4, did not clearly show the significant differences in the results, discuss the result of the assay properly in the results and discussion. Also, compare your data with the results of other researchers on the antioxidant activity of the flavonoids that were tested.
· Table 1. It should be Code instead of Abbr. LOD and LOQ, not LODs and LOQs., Just R2 since it's already understood, and should be Recover (%). Also, Improve the Title of the Table.
· Table 2 is missing. And should be included in the main manuscript and not as supplementary material.
· Table 2. Specify the B and C numbers
· Table 2. It is difficult to understand whether the values shown and reported here are expressed in % inhibition or as concentration. Maybe fix your labels properly
· Table 2. Label properly. For the first label, it should only be PAH Compound since there is no value for concentration, next, the abbreviations should be labelled as Code, A separate Column that shows the % inhibition should be added for each flavonoid for better understanding. Each label should be in ug/kg and in % Inhibition for each flavonoid.
· Table 2. Footnotes should be labeled as Legend.
· Section 3.2. Divide this subsection into the GS-MS Validation analysis, and then the inhibition action against PAH.
· Note: uq/mL was used for the validation and ug/kg was used for the analysis. Can you try to have a consistent unit?
· The results shown in Table 2 lacked discussion and comparison with other reports. Please try to address and improve this flaw.
· A figure showing a chromatogram of a particular PAH identified in a meat system and inhibited by a flavonoid should be included as it will provide a better perspective
· The discussion of the results in Sections 3.3 to 3.5 would be more meaningful if discussed with p values and statistical significance in which you had performed. Also, as much as possible, relate your data with the data reported by other researchers.
· You have explained the difference between the hydroxylation groups but did not delve into the difference in the reactivity between the different types of flavonoids used, in particular, taxifolin is a flavanonol, while the rest of the flavonoids used are flavanols. Address this issue because this can explain why you have seen a negative correlation for FRAP and DPPH.
· In figure 4 and section 3.6, take note that the lower the EC50, the better is the inhibition. Is this what you had in mind when you performed the correlation experiments?
· The discussion is lacking and should be improved. Furthermore, the significance of this research in meat processing or preservation is lacking in the discussion which weakens the paper.
Conclusion:
· Should be brief and should be based strictly on the findings of the research.
Reviewer 2 Report
Comments and Suggestions for Authors
The major objective of the manuscript was to explore the relationship between the structure of several flavonoids and their ability to inhibit PAHs.
Please consider the following suggestions:
Please do not use abbreviations in the title of the manuscript.
Please provide the conclusion of your study at the end of the Abstract.
Line 32. Please insert the full names of all the compounds in the PAH8 group.
Figure 1. Please use numbers of letters to indicate the position of the chemical groups indicated in the table presented at the bottom of the figure.
Please insert a chromatogram that shows clearly all the compounds that you determined (e.g., addressing the overlapping)
Please insert table 2 near the text in which you refer to it.
The references used in the manuscript are not recent. Please search the scientific literature and insert more recent articles.
Comments on the Quality of English Language
Minor editing of English language required
Reviewer 3 Report
Comments and Suggestions for Authors
The present original article submitted for publication deals with an extremely relevant topic in view of the consumers demand for healthy and safe foods. Despite this, I believe that this work should be subject to major revisions.
The issues to be addressed are listed below:
- Line 32: the authors should explain the abbreviations standing for BaA, Chry, BbF, BkF, BaP, InP, DahA, and BghiP
- Lines 45, 58, 61, 64, 69: the authors repeat several times “relationship between”, similar expressions could be used alternatively
- Line 55: Figure 1 does not illustrate the referred flavonoids categories. Indeed, the authors do not indicate to which category the tested flavonoids belong. As such, the figure should be improved or the figure citation removed from this sentence
- Line 75: the figure should be improved due to the 1) poor quality of the images, 2) incomplete names of some structures (e.g. taxifolin), 3) the columns of the table in the lower part of the figure are not properly identified
- Line 81: Grammatic and phrasing in the section “Materials and Methods” should be improved
- Line 92: Where it is “benzo(ghi)perylene” it should be “benzo(g,h,i)perylene”
- Lines 92 and 96: “QuEChERS kits were obtained from Supelco (PA, USA)” is repeated
- Line 95: Where it is “all solvent high-performance liquid chromatography (HPLC) grade— acetonitrile, methanol (HPLC grade)”, it should be ”all solvent high-performance liquid chromatography (HPLC) grade (acetonitrile and methanol)“
- Line 102: the authors do not characterize the flavonoids extract. Ethanol is mentioned, but only in section 2.4. For this reason, details regarding the solvents used and compounds concentrations should be provided
- Line 104: “10 s” are separated in two lines
- Line 105: where it is “absorbance was 517 nm” it should be “absorbance was measured at 517 nm”
- Line 107: where it is “EC50” it should be “EC50”. Revise the entire manuscript as similar problems are found in other parts of the text
- Line 122 to 132: units should be separated from the number
- Lines 88 and 130: the sentence “was purchased from a local market” is repeated
- Lines 130 to 132: the ground beef was cut in smaller pieces before lyophilization? Considering a future transfer of the proposed technology for food industry it is not clear the reason why the authors have freeze-dried the meat, neither the option for an oil bath. Did the authors considered to test typical food processing conditions like grilling and roasting? These options should be clearly explained.
- Line 147: did the authors use any homogenizer? Or do they mean “mixed” instead of “homogenized”
- Line 158: instead of “Cannada” it should be “Canada”
- Line 161: instead of “1.5 mL min-1” it should be “1.5 mL/min”
- Line 176: details regarding Pearson correlation should be indicated here as well
- Line 183: the section corresponding to the “Results and Discussion” should be globally improved. For example, more details on results regarding antioxidant activity (with values), currently in section 3.1, should be provided. Then the authors discuss LOD and LOQ in the section regarding the inhibitory action of antioxidants over PAH formation, while the most relevant discussion regarding PAHs is disperse in sections 3.3, 3.4 and 3.5. These seems quite confusing. In my opinion, and considering the relation the authors want to stablish between antioxidant activity and PAHs formation, the best option would be to discuss them together. This could be achieved if the results presentation and the discussion are presented in independent sections.
- Line 207: bars represent standard deviation? Please include this information in the figure caption.
- Line 210 to 215: it seems that the peaks retention time presented in Figure 3 and Table 1 do not match. For example, BaA and Chry co-elute in Figure 3, yet their retention times, according to Table 1, correspond to 10.39 and 10.41, respectively. In turn, BbF and BkF that are perfectly separated in the chromatogram show with retention times of 11.39 and 11.40 (according Table 1). Please confirm that the figure and table correspond to the same separation method and that the retention times included in the table are correct.
- Line 219: authors never compare their results with current regulations. It would be interesting to demonstrate that the contamination levels comply with the current regulations. Are there any Thai regulations for PAHs maximum levels? Despite of this, and considering authors used European regulation to support recoveries deviations, they could also use it in PAHs results discussion. It would be also interesting to provide results for PAH4 (included in current European Regulation: 2023/915 from 25 of April 2023)
- Line 228: Table 2 should be closer to its citation in the text
- Line 270: please identify inside brackets the light PAHs (BaA and Chry)
- Line 302: where it is “lower, by 33% and 57%, compared to quercetin samples” it should be “lower (33%) compared to quercetin samples (57%)”
- Lina 309: authors should explain in the text why they use “r” and “R2” to compare the correlations and include this information in the section 2.7 as well.
Comments on the Quality of English Language
Please check the section Comments and Suggestions for Author.
Reviewer 4 Report
Comments and Suggestions for Authors
1. In this study authors investigated how flavonoids can interrupt the formation of PAHs in a heated meat model. The manuscript is well organized. The major drawback is the discussion part in which it is only discussed how the antioxidant activity of flavonoids is related to their structure, which is a well known issue among the scientific community.
Please find below my comments and suggestions
1. In figure 1 the position of hydroxyl groups of the taxifolin structure, is hidden.
2. In figure 1 which is the meaning of + and – symbols? In addition, at the “table” of figure 1 why rutin is presented as rutin trihydrate? Finally, you should add a column heading in this “table”, that will identify each column
3. Lines 105-106 page 4: “Finally, the mixture's absorbance was 517 nm”. Please correct to “the mixture absorbance was read at 517 nm.” Same at line 117.
4. Lines 115-116: “Methanol was used to dilute the ABTS•+ solution (1 mL) to 0.700 ± 0.025 at 734 nm (50 mL).” Those volumes of 1mL and 50 mL are not clear. Please explain better
5. Line 107 page 4: “The EC50 is the sample required to reduce DPPH absorbance by 50%.”. The EC50 is not the sample, is the concentration of the sample required to reduce DPPH absorbance
6. Usually, a standard curve with a strong antioxidant accompanies experiments of DPPH and ABTS assays. You performed experiments to construct a standard curve for FRAP assay. Why didn’t you perform similar experiments for DPPH and ABTS assays?
7. Lines 149, 152, page 5: Please write the chemical formulas MgSO4 and CH3COONa properly. Please correct all the chemical formulas
8. Line 161, page 5: Please correct the “1.5 mL min-1” to 1.5 mL min-1
9. Line 178, page 5: please rephrase the following sentence “the average results were reported along with the standard deviation (SD).” The meaning is understood; however, syntax errors exist.
10. Lines 190-191, page 5: you should give the EC50 and EC1 values also in the text. In addition, you should specify whether the final concentration of the flavonoids used to interact with the free radicals is the same.
11. Lines 280-281, page 9: “This was evident in EC50 values of 29.68 mg/L for DPPH and ABTS…”.The EC50 value of 29.68 mg/mL is equal for both assays? Please check
12. Lines 282-285: please check again this sentence. The meaning is not clear
General comments:
1. The discussion is only an interpretation of your results, which is based on quite known information (structure-activity relationship and antioxidant activity of flavonoids). Comparison with similar works is missing. You should enrich the discussion part.
2. Please correct grammatical errors and syntax errors: see for example lines 106; 185; 287; 289
Comments on the Quality of English Language
Minor editing is required
Round 2
Reviewer 1 Report
Comments and Suggestions for Authors
Dear Aurhors,
Well done in improving your paper.
I have few comments and suggestions that require your attention:
Introduction:
· Table 1, should be The molecular structures of the tested flavonoids
· Table 1, the proposed structure do not conform for taxifolin. I suggest you show another flavonoid backbone for taxifolin.
· Table 1, Place Flavonoid in the center of the cell.
· Table 1, what does ** mean after the molecular formula? Remove it if its not necessary.
· Table 1, indicate the source or how the structure was generated
Materials and Methods:
· Section 2.1. should be Auckland
· Page 3, Line 124-125. It should be in a Section entitled Sample Preparation or can be included in the materials and methods. Furthermore, improve the English of this paragraph.
· No need to repeat the model of the UV-VIS spectrophotometer (Model: Genesys 10-S, USA) if you had already indicated it in the first section.
· Page 4, Line 162… should be The final result
·
Results and Discussion
· Page 5, Line 220, should be: is shown in
· Table 3. There is a ug/kg unit floating near the structures. Remove it and instead, include it after the EC50 for ABTS and DPPH, and EC1 for FRAP.
Comments on the Quality of English Language
The authors need to re read the paper once again to check for grammatical errors and to improve the cohesiveness of the sentences in the paragraphs.
Reviewer 2 Report
Comments and Suggestions for Authors
The manuscript was improved as suggested.
Comments on the Quality of English LanguageMinor editing of English language required
Author Response
The manuscript was improved as suggested.
Minor editing of English language required.
The manuscript has been thoroughly reviewed for grammar and spelling errors.
Reviewer 3 Report
Comments and Suggestions for Authors
The authors have addressed most issues raised in my first report. Yet, for me is not entirely clear their option for the freeze-dried meat. In their response they refer: ”The present study aimed to investigate the inhibitory effects of flavonoids, heated meat model system was used to limited other external factors.” Could you please detail what you mean by “other external factors”? I also consider that these explanations should be included in the manuscript in order to support the rational for the set experimental design.
In line 295 the units regarding the antioxidant activity are missing.
Reviewer 4 Report
Comments and Suggestions for Authors
The manuscript has been significantly improved, and I appreciate the authors' efforts to respond to my comments. Some minor revisions listed below
1. What does the asterisk(s) indicate in tables 1 and 2? Please give the explanation
2. In table 3 please explain the letters a-d
3. Please check throughout the text for syntax errors or misspelling words for example
‣line 220, please correct to shown
‣line 274, please delete the word “that”, it is written twice
‣line 331, syntax errors
Comments on the Quality of English Language
Minor editing of the english language is required
